# Differential Response and Recovery Dynamics of HSPC Populations Following *Plasmodium chabaudi* Infection

**DOI:** 10.3390/ijms26062816

**Published:** 2025-03-20

**Authors:** Federica Bruno, Christiana Georgiou, Deirdre Cunningham, Lucy Bett, Marine A. Secchi, Samantha Atkinson, Sara González Antón, Flora Birch, Jean Langhorne, Cristina Lo Celso

**Affiliations:** 1Department of Life Sciences, Imperial College London, London SW10 9NH, UK; federica.bruno21@imperial.ac.uk (F.B.);; 2Sir Francis Crick Institute, London NW1 1AT, UK; 3Centre for Haematology, Department of Immunology and Inflammation, Imperial College London, London W12 0NN, UK

**Keywords:** malaria, hematopoietic stem and progenitor cells, recovery from infection

## Abstract

Severe infections such as malaria are on the rise worldwide, driven by both climate change and increasing drug resistance. It is therefore paramount that we better understand how the host responds to severe infection. Hematopoiesis is particularly of interest in this context because hematopoietic stem and progenitor cells (HSPCs) maintain the turnover of all blood cells, including all immune cells. Severe infections have been widely acknowledged to affect HSPCs; however, this disruption has been mainly studied during the acute phase, and the process and level of HSPC recovery remain understudied. Using a self-resolving model of natural rodent malaria, infection by *Plasmodium chabaudi*, here we systematically assess phenotypically defined HSPCs’ acute response and recovery upon pathogen clearance. We demonstrate that during the acute phase of infection the most quiescent and functional stem cells are depleted, multipotent progenitor compartments are drastically enlarged, and oligopotent progenitors virtually disappear, underpinned by dramatic, population-specific and sometimes unexpected changes in proliferation rates. HSPC populations return to homeostatic size and proliferation rate again through specific patterns of recovery. Overall, our data demonstrate that HSPC populations adopt different responses to cope with severe infection and suggest that the ability to adjust proliferative capacity becomes more restricted as differentiation progresses.

## 1. Introduction

Mature blood and immune cells are continuously replenished through a dynamic process known as hematopoiesis. During homeostasis, this process is tightly regulated and characterized by a hierarchical cellular differentiation system so that the high turnover of blood cells is maintained lifelong. At the top of the hematopoietic system, a pool of hematopoietic stem cells (HSCs) is able to generate all differentiated blood cell types by giving rise to several different progenitors thanks to their self-renewal and multilineage differentiation capacity [1]. It is now clear that while, usually, most HSCs are in a quiescent state, the massive daily turnover of blood cells is sustained by increasingly proliferative pools of progenitors, from multipotent progenitors (MPPs), direct progeny of HSCs, to more mature, oligopotent progenitors [2]. A wealth of studies have characterized hematopoietic stem and progenitor cell (HSPC) populations [3]. Of note, self-renewal and multilineage potential have been shown to decrease alongside the lineage commitment process. While MPP populations are lineage-biased but retain multipotentiality, common myeloid progenitors (CMPs), granulocyte–macrophage progenitors (GMPs), megakaryocyte–erythrocyte progenitors (MEPs), and common lymphoid progenitors (CLPs) are committed to the lineages indicated by their names [4]. Overall, decades of research have shown that these populations differ in terms of proliferation kinetics, homeostatic functions, maintenance mechanisms, and lifespans, highlighting the high heterogeneity present within the hematopoietic system and even within the HSC pool itself.

During hematopoietic stress, the dynamics of this tightly regulated but flexible system change, leading to disrupted hematopoiesis. In response to several models of infections, HSCs are forced out of quiescence in order to cope with the increased demand for immune cells, which are consumed during the fight against the pathogen in question [5,6,7]. Collectively, HSPCs have been shown to sense and respond to cytokines and pro-inflammatory stimuli, resulting in several phenomena such as increased proliferation and differentiation towards the myeloid lineage, leading to emergency myelopoiesis, as well as increased mobilization outside of the bone marrow (BM) [8,9]. Many studies reported that forced entry into the cell cycle resulted in replication stress within HSCs, leading to their loss of functionality, demonstrated by a decrease in repopulation potential in transplantation assays [10,11]. Following transient infections, HSCs have been shown to return to a dormant state while continuous stimulation to proliferate in response to chronic inflammation consequently has led to accelerated stem cell exhaustion and long-term functional impairment [12,13,14,15,16,17]. In the latter, the ability of some HSCs to return to quiescence has been proposed to be critical to retain some function [11].

Malaria is a severe and life-threatening disease and is still a leading cause of mortality. Of the more than 240,000 million cases of malaria in humans per year, most are resolved; however, there are more than 600,000 deaths per year, 90% of which are caused by *Plasmodium falciparum* in children in sub-Saharan Africa [18]. Following a bite of the female *Anopheles* mosquito, *Plasmodium* sporozoites enter the host and travel via the lymphatic system from the skin to the liver, invading hepatocytes. Here, the parasites multiply asexually and mature into schizonts, which eventually rupture the hepatocytes leading to the release of merozoites into the bloodstream. In the blood, the parasites invade red blood cells and perform asexual replication, causing the hemolysis of infected RBCs and release of more merozoites, leading to the onset of clinical symptoms in the host [19]. Malaria has been reported to compromise the immune system both in the short term, leading to complications such as cerebral malaria, acute respiratory distress, and kidney dysfunction and, in the long term, resulting in severe anemia and the loss of BM cellularity [20,21,22,23]. The pathophysiology of malaria is centered around acute systemic inflammation, which results in pro-inflammatory cytokine production, lymphocyte activation, and vessel congestion by the adhesion of infected RBCs to the endothelium [20,21,22,23,24]. Recently, parasites have also been documented to circulate, reside, and infect cells in the BM, suggesting the BM microenvironment as a potential *Plasmodium* reservoir [20,21,22,23]. Acute *Plasmodium* infection has been proven to alter hematopoiesis; for instance, a *Plasmodium*-induced IL-7R^+^ c-Kit^hi^ myeloid-primed progenitor population was reported to aid in the clearance of infected RBCs [25]. Analyses of primitive HSPC populations in mice infected by *Plasmodium berghei* highlighted emergency myelopoiesis at the expense of erythroid and lymphoid lineages, the loss of functional HSCs, and a role for the BM microenvironment in mediating such damage and as a possible therapeutic target to preserve HSC function during the acute phase of the infection [7,26]. The extent and dynamics of HSPC recovery following malaria and other severe yet non-fatal infections have remained understudied despite holding clues on potential strategies to alleviate both short and long-term consequences of infection. These are particularly important for malaria because the severe forms of this disease tend to occur in children. Generally, experimental strong inflammatory stresses have been linked to loss of HSC function, HSC ageing, and the selection of HSC clones carrying mutations linked to clonal hematopoiesis and, eventually, malignant transformation [10,15]. However, less is known about recovery from physiological severe infections.

While *P. berghei* is an excellent model of severe and cerebral malaria, it is fatal to mice unless treated with drugs that are themselves likely to affect hematopoiesis. Here we employ a different model of rodent malaria, *Plasmodium chabaudi chabaudi*, from which mice can recover without any pharmacological intervention. Through a time course analysis, we systematically characterize the effects of *P. chabaudi* on primitive HSPCs (HSCs and all MPP populations) and downstream oligopotent progenitor (CMP, GMP, MEP, and CLP) populations in the BM throughout the acute and recovery stages. Immunophenotypic and EdU incorporation assays allow us to demonstrate that while, eventually, hematopoiesis returns to homeostatic conditions, HSPC populations are heterogenous in their response and recovery dynamics. These findings have important implications for better harnessing hematopoiesis to enable both infection resolution and healthy ageing.

## 2. Results

### 2.1. Plasmodium chabaudi Infection Is an Effective Model to Study Perturbed Hematopoiesis and Its Recovery

To investigate the responses of hematopoietic cells and their progenitors during acute infection and following resolution, C57BL/6 wild-type mice were either mock-injected or infected with *Plasmodium chabaudi AS*, and their peripheral blood (PB) and BM were analyzed at different time points post infection (p.i.). (Figure 1a). *P. chabaudi* acute infection was characterized by peak parasitemia at day 11 p.i., which was then resolved to subpatent levels by day 24 p.i. (Figure 1b and Appendix A), apart from a small patent recrudescence at day 29 p.i., consistent with previous observations [27,28,29]. Using parasitemia levels as a guideline, we defined different stages of infection and selected appropriate time points to investigate specific responses associated with different phases of infection. Namely, PB and BM samples from control and infected mice were analyzed at day 8 (early response: pre-parasitemia peak), at days 11 and 15 (acute phase: peak and post-peak of parasitemia, respectively) and days 24, 29, and 60 (recovery phase: post-peak, and very-low-to-undetectable parasitemia) p.i. (Figure 1a,b and Appendix A).

Acute infection was accompanied by anemia as shown by a significant decrease in the counts of RBCs in the PB at day 11, which gradually recovered and returned to normal levels by day 29 p.i. (Figure 1c, left panel). We observed different dynamics for white blood cells (WBCs): while they significantly decreased at day 11 p.i., they recovered faster than RBCs and overall presented larger variability within the cohort. Interestingly, at day 15 p.i., WBCs showed a trend to increase, which would be consistent with the activation of the immune system elicited by *P. chabaudi* [20] (Figure 1c, middle panel). Lastly, platelet counts were largely stable throughout the infection, with only a small but significant decrease in platelet numbers in infected animals at day 15 p.i. (Figure 1c, right panel). The major cytokines produced in response to *Plasmodium* infection are IFN-γ and TFN-α, and both were elevated in the serum of infected mice at day 11p.i. and returned to control levels at day 15 p.i. (Appendix A).

To validate that the acute phase of *P. chabaudi* infection is associated with perturbed hematopoiesis, we analyzed the BM compositions of the same mice using flow cytometry. First, we observed broad changes in the overall primitive HSPC and oligopotent progenitor compartments, where the first compartment includes HSCs and MPP populations and the second one includes the common lymphoid progenitor and bipotent myeloid progenitors (Appendix A and Figure 1d). At the peak of infection (day 11 p.i.), we observed an expansion of the LKS (Lineage^−^c-Kit^+^Sca-1^+^ cells) compartment, which comprises MPPs and HSCs, in agreement with previous studies [30] (Figure 1d,e). This infection-induced increase in LKS cells’ absolute number was still observable at day 15 p.i., albeit at lower levels, and LKS cell counts had returned to control levels during early recovery (day 24 p.i.). The changes in the LKS population were mirrored by a trend for the contraction and recovery of the more differentiated, myeloid-primed LK (Lineage^−^c-Kit^+^Sca-1^−^) compartment, which was not statistically significant due to higher variability in the size of this population (Figure 1f). However, when the whole Lineage^−^ c-Kit^+^ compartment (LK and LKS cells combined) was considered, the proportion of LKS cells within the compartment was strikingly increased, despite it normally being the smaller component (Figure 1d,g). For both LKS and LK compartments, counts recovered by day 24 and remained essentially unchanged from then (Appendix A). Taken together, these data show that *P. chabaudi* is an effective model to study changes in HSPC populations in response to an acute infection and during recovery from it.

### 2.2. Infection-Induced Alterations in the Size of Primitive Hematopoietic Stem and Progenitor Populations Are Resolved upon Pathogen Clearance

To investigate *P. chabaudi* infection-induced changes in primitive hematopoietic stem and progenitor cells, we examined subpopulations within the LKS compartment by employing signaling lymphocyte activation molecule (SLAM: CD150 and CD48) and FLK2 (CD135/Flt3) markers. In this paper, we define MPP4 as LKS Flk2^+^ CD150^−^ cells and MPP3 and MPP2 as LKS Flk2^−^ CD150^−^ CD48^+^ and LKS Flk2^−^ CD150^+^ CD48^+^, respectively, while MPP5-6 was defined as LKS Flk2^−^ CD150^−^ CD48^low/−^, according to previous studies (Figure 2a) [3,31,32]. Lastly, we analyzed, separately, the more quiescent and primitive CD48^−^ HSC population, defined as LKS Flk2^−^ CD150^+^ CD48^−^, and the more proliferative, less primitive CD48^low/−^ HSPCs (LKS Flk2^−^CD150^+^ CD48^low/−^), as we described previously [33] (Figure 2a and Appendix A). Consistent with the hallmarks of multiple inflammatory models [5,7,12,26], *P. chabaudi* caused a dramatic swelling of all MPP populations, which contributed to the overall LKS population expansion observed earlier, while the most primitive CD48^−^ HSCs appeared reduced (Figure 2a). Overall, populations returned to levels similar to controls by day 24 p.i., followed by a small enlargement of MPP populations at day 29 p.i., which was then recovered at day 60 p.i. (Figure 2a and Appendix A). To test whether the strength of infection may affect the degree of primitive hematopoietic progenitors’ responses, we additionally analyzed a more severe model of rodent malaria, *Plasmodium chabaudi CB* [34] (Appendix A). Specifically, while *P. chabaudi CB* is associated with higher levels of parasitemia and a decreased probability of survival compared to *P. chabaudi AS* (Appendix A), the expansion of LKS and MPP populations and their subsequent recovery were largely similar in these two models (Appendix A). Interestingly, while in *P. chabaudi AS* we only observed a trend for HSC numbers to increase during the recovery phase, in *P. chabaudi CB* this trend was significant (Appendix A). Taken together, these data highlight how a more severe rodent malaria model leads to similar effects on primitive HSPC populations.

To quantify the precise effect of *P. chabaudi* infection on all the above-mentioned populations, we calculated and analyzed their absolute numbers (Figure 2b–g). The absolute number of CD48^−^ HSCs halved compared to controls during the acute phase of infection (days 11 and 15 p.i.) and showed a highly variable rebound phase, with a trend to increase at day 24 p.i., although this was not statistically significant. By day 60 p.i., CD48^−^ HSC numbers had returned to control levels (Figure 2b). Instead, CD48^low/−^ HSPC numbers were moderately but significantly increased at day 11 p.i., and equivalent to control levels on all other days, except for a trend towards an increase on day 29 p.i. (Figure 2c). The myeloid/lymphoid-primed MPP5-6 cells were quite stable, with only a non-significant trend to increase at day 11 p.i. (Figure 2d). As expected, the myeloid-primed MPP populations, namely MPP2 and MPP3, had a massive increase at the peak of infection and subsequently recovered. Specifically, they were 50 and 40 times more numerous compared to the control group at day 11 p.i., respectively, and returned to normal levels by day 24 (Figure 2e,f). Lymphoid-primed MPP4 cells showed a smaller, 6-fold increase in population size at day 11 p.i., which was followed by a quicker recovery, with baseline numbers reached by day 15 p.i. (Figure 2g). These data highlight that *P. chabaudi* infection-induced emergency granulopoiesis is fueled by a dramatic increase in the number of myeloid-primed MPPs and by a decrease in the number of quiescent HSCs. Although the sizes of these populations recover upon pathogen disappearance, the question remains regarding the proliferation dynamics that underpin the changes observed during the acute and recovery phases of infection.

### 2.3. Heterogeneous Proliferation Dynamics Within the Primitive Haematopoietic Compartment During and Following Infection

To shed light on possible mechanisms underlying the increases and decreases in primitive hematopoietic cell numbers, we assessed the proliferation of these populations by measuring in vivo EdU incorporation. EdU is a thymidine analog that intercalates in the DNA of cells during the S-phase of the cell cycle and can be detected ex vivo using flow cytometry. To gain a quantitative and comprehensive understating of proliferation changes in HSPCs, we assessed both the absolute number and percentage of proliferating (EdU^+^) cells within each primitive population (Figure 3 and Appendix A). While the absolute numbers are useful to have an overall quantification, percentages are key to understand whether increases or decreases in the absolute numbers of cell populations result from increases or decreases in their proliferation rates. Because CD48^−^ HSCs are very rare and quiescent, flow cytometry data were not sufficiently robust to quantify the absolute number of proliferating cells. However, we consistently detected more numerous EdU^+^ cells in this population at days 11 and 15, and not at any other time point analyzed (Figure 3a and Appendix A). Consistent with this, we detected a significant increase in the percentage of EdU^+^ CD48^−^ HSCs during the acute phase of infection, which returned to control levels by day 24 p.i. (Figure 3b). Similarly, both the absolute number and percentage of proliferating cells within CD48^low/−^ HSPCs were increased at the infection peak and returned to control levels by day 15 p.i and day 24 p.i., respectively (Figure 3c).

On the other hand, MPP populations presented more heterogeneous kinetics (Figure 3d–g). Interestingly, while the absolute numbers of MPP5-6 only showed a trend to increase (Figure 2d), the number of EdU^+^ cells in this population was 35 times higher in infected animals compared to control animals at day 11 p.i. (Figure 3d, left panel; Appendix A). Similarly, percentages of EdU^+^ cells within MPP5-6 increased from 2.8% in control to 42% at day 11 p.i. (Figure 3d, right panel). However, while numbers recovered quickly, already by day 15 p.i., percentages of proliferating cells within these populations returned to control levels only at day 24 p.i. (Figure 3d, right panel). When analyzing MPP2, 3, and 4 populations, we noticed that the expansion of cell numbers observed during the acute phase of infection (Figure 2e–g) was associated with a dramatic increase in the absolute number of EdU^+^ cells within all the populations, but with cell-type-specific dynamics in the proportion of EdU^+^ cells and their time of recovery (Figure 3e–g and Appendix A). In detail, the massive expansion observed in megakaryocytic/erythroid-biased MPP2 at the peak of infection (Figure 2e) was accompanied by a 68-fold increase in the numbers of proliferating cells within this compartment, which recovered by day 24 p.i. (Figure 3e, left panel). Interestingly, no changes were observed in terms of percentages of proliferating cells within MPP2 throughout the infection (Figure 3e, right panel). Similarly, at day 11 p.i., the enlargement of the myeloid-biased MPP3 compartment was also accompanied by a massive increase in the absolute numbers of proliferating cells within this population compared to controls (120 times higher), which then returned to homeostatic levels by day 24 p.i. (Figure 3f, left panel).The proportion of EdU^+^ MPP3 cells also increased, though only about 2.5-fold, and recovered already by day 15 p.i. (Figure 3f, right panel). Lastly, while lymphoid-biased MPP4 cells showed a 6-fold population enlargement at day 11 p.i. (Figure 2e), absolute numbers of proliferating cells within this compartment were 12-fold higher compared to controls at day 11 p.i., and returned to control levels by day 15 p.i. (Figure 3g, left panel). Similarly, percentages of proliferating cells within the MPP4 population also peaked at day 11 p.i. and were back to homeostasis at day 15 p.i. (Figure 3g, right panel). Taken together, these data demonstrate that *P. chabaudi* infection leads to a significant but transient increase in the number of proliferating cells within the primitive hematopoietic compartments, including for CD48^−^ HSCs, which is then followed by recovery dynamics specific for each population.

### 2.4. P. chabaudi Infection Dramatically Affects Oligopotent Progenitor Numbers

To gain a fuller picture of the changes occurring in different hematopoietic progenitor populations in response to malaria infection, we extended our flow cytometry analysis and investigated the numbers and proliferative states of MPPs’ immediate progeny. These populations are known to be lineage-restricted and therefore oligopotent and more proliferative than primitive HSPCs [35]. Guided by well-established definitions based on the expression of CD34 and CD16/32 receptors, we defined CMPs, GMPs, and MEPs within the LK compartment as CD34^+^ CD16/32^−^, CD34^+^ CD16/32^+^, and CD34^−^ CD16/32^−^, respectively (Figure 4a and Appendix A) [36]. Lastly, CLPs were defined as Lin^−^ c-Kit^int^ Flk2^+^ CD127^+^ (Figure 4b and Appendix A) [7]. We observed a drastic decrease in the CMP, GMP, MEP, and CLP populations at the peak of the infection, when parasitemia was around 25% (Figure 1b), and again population-specific patterns of recovery emerged at the later time points (Figure 4).

Specifically, CMP cells’ absolute numbers decreased to nearly zero at day 11 p.i, were still reduced compared to controls at day 15, and returned to homeostatic levels only by day 29 p.i. (Figure 4c and Appendix A). GMP and MEP numbers in infected mice were reduced by 75% at day 11 p.i. compared to control mice; however, recovery was already observed at day 15 p.i. for MEPs, while it was reached by day 24 p.i. for GMPs (Figure 4d,e). Interestingly, despite CLP numbers being reduced to nearly zero at day 11 p.i., control levels were quickly re-established by day 15 p.i. (Figure 4f). Additionally, we included the LK CD34^−^CD16/32^+^ population in our analysis, which was negligible in the steady state, but we noticed that it increased during infection (Figure 4g). This population has been generally neglected and is therefore poorly characterized. The size of this population increased about 2-fold only by day 15 p.i. and returned to steady state size by day 24 p.i. (Figure 4g). Overall, these data demonstrate that despite a general loss of oligopotent progenitors within the BM compartment during *P. chabaudi* infection, their numbers are recovered within one month p.i., with the CMP population being the slowest to recover and thus being the most affected.

### 2.5. Oligopotent Progenitors’ Proliferation Is Minimally Affected During P. chabaudi Infection

To explore how mature progenitors’ numbers respond to and recover following *P. chabaudi* infection, we investigated their proliferation levels throughout infection and recovery. Similar to our analysis of primitive HSPCs, we assessed both the absolute number and percentage of proliferating cells within each population, building a comprehensive overview of the dynamics taking place at this stage of hematopoietic development (Figure 5 and Appendix A). Predictably, the overall decrease in the absolute numbers of mature progenitors was accompanied by an overall decrease in the absolute numbers of proliferating cells within most subpopulations (Figure 5a–e, left panels). Surprisingly, the reduction in proliferating cells’ numbers was not associated with either a reduction or an increase in the proportion of proliferative cells in any of the populations as, for the most part, we did not observe any significant change in the percentages of proliferating cells within the subpopulations (Figure 5a–e, right panels). As expected, in homeostasis (control mice), we found that most mature progenitors, namely GMPs, MEPs, and CD34^−^ CD16/32^+^ cells, were associated with a very high proliferative index as approximately 40–50% of cells within each population were in the S-phase (Figure 5b,c,e, right panels). Interestingly, CMPs and CLPs were found to have a lower, but still relatively high, proliferative index, with an average of 20% of proliferating cells within each population (Figure 5a,d, right panels).

A detailed analysis of the dynamics of each progenitor population uncovered, again, specific responses. Due to the virtual disappearance of the CMP population at day 11 p.i., their proliferation could not be assessed on that day. However, absolute numbers of proliferating cells within CMP were decreased on days 15 and 24 p.i., recovering by day 29 p.i. (Figure 5a, left panel, Appendix A). Predictably, this recovery was in line with overall absolute numbers of CMPs reaching control levels by day 29 p.i. (Figure 4c). Instead, percentages of proliferating cells within this population remained similar to those of control mice throughout the infection (Figure 5a, right panel). Similarly to those of CMPs, numbers of proliferating GMPs also decreased at acute infection, recovering at day 29 p.i. (Figure 5b, left panel, Appendix A). Of note, percentages of proliferating GMPs increased at day 11 p.i. but were similar to those of GMPs from control animals for all other time points analyzed (Figure 5b, right panel). Next, absolute numbers of proliferating MEPs decreased at day 11 p.i. (Figure 5c, left panel), coupled with a decrease in the proportion of proliferating cells within the remaining population (Figure 5c, right panel). MEP numbers and proportions of proliferative cells were similar to those of controls for all other time points (Figure 5c). Interestingly, with regards to CLPs, the absolute number of proliferating cells within this population was lower than control only at day 24 p.i. (day 11 was not included as CLPs were virtually absent at this time point) (Figure 5d, left panel). Similar to the other populations, the percentage of proliferating cells did not show any alteration (Figure 5d, right panel). Strikingly, when investigating the CD34^−^ CD16/32^+^ population, we observed a very different response as there was a trend to increase in the number of proliferating cells at the peak of parasitemia, day 11 p.i., with high mouse-to-mouse variability, followed by a significant increase compared to control at the first time point post parasitemia peak, day 15 p.i., and a quick recovery, already achieved by day 24 p.i. (Figure 5e, left panel). However, no changes were detected in the percentage of proliferating cells throughout our analysis (Figure 5e, right panel). Taken together, these data demonstrate that committed hematopoietic progenitor populations maintain a relatively steady proliferative rate in response to *P. chabaudi* infection and subsequent recovery.

### 2.6. Primitive and Committed Hematopoietic Progenitor Populations Respond to and Recover from P. chabaudi Infection Differently

To gain a system-level understanding of the cellular dynamics driving the hematopoietic response to and recovery from *P. chabaudi* infection, we combined all our observations in summary graphs where we could more effectively compare population dynamics (Figure 6). This analysis highlighted stark differences in the response of different cell populations. Within the primitive hematopoietic compartment, CD48^−^ HSCs were the only population with a significant reduction in size at day 11 and 15 p.i., and their downstream progenitor populations showed increases ranging from minimal to very dramatic in size at day 11 p.i., with MPP 2 and 3 taking longer to return to control size (Figure 6a). Despite no statistically significant differences being highlighted for any of these populations after day 15 p.i., it was striking that all populations were synchronized in showing a trend to increase at day 29 p.i., likely due to the small increase in parasitemia, and again steady-state sizes at day 60 p.i. (Figure 6a). The overview of proliferation dynamics highlighted how the increase in MPP numbers was fueled by increased proliferation in all populations, albeit again at different degrees, and with the most primitive population, CD48^−^ HSCs, showing one of the highest increases in proliferation, likely to fuel the activation of the hematopoietic cascade and regeneration of downstream progenitors (Figure 6b). Interestingly, while percentages of proliferating cells for most MPPs peaked with parasitemia at day 11 p.i., CD48^−^ HSCs had the highest proportion of EdU^+^ cells at day 15 p.i. (Figure 6b).

On the other hand, downstream oligopotent, committed progenitors presented a very different picture, with most populations massively reducing in size at the peak of parasitemia (day 11 p.i.), starting to recover steadily by day 15 p.i., and showing virtually no rebound in numbers at later time points (Figure 6c). The grouped analysis confirmed the overall lack of dramatic changes in proliferation for all these populations, the only exception being MEPs, which showed a trend towards reduced proliferation at day 11 p.i. (Figure 6d). All these data together demonstrated how different populations within the hematopoietic tree adopt very different dynamics when responding to and recovering from infection. Generally, while primitive progenitors massively increased both their population sizes as well as proliferation levels, mature progenitor populations were largely affected through a high decrease in size but no significant changes in proliferation rate. Moreover, quiescent HSCs were shown to respond to infection by decreasing in number at peak infection followed by an increase in the percentages of proliferating cells. For the most part, recovery, in terms of population size and proliferation, was observed within one month p.i.

## 3. Discussion

Many studies have described the effects of acute infection on the primitive compartment of the hematopoietic system, including an increase in proliferation, mobilization, and myeloid differentiation. However, many questions remain unanswered regarding the recovery of the hematopoietic system after inflammatory stimuli, including when and to what extent recovery occurs. These questions are crucial to address, particularly for individuals living in endemic areas who are likely to experience repeated infections such as recurring malaria. For this study, we employed a non-lethal model of rodent malaria, infection by *P. chabaudi*, from which mice can recover without pharmacological interventions. This is ideal to characterize the dynamics of different hematopoietic populations during the acute and recovery phases of blood-stage malaria infection without any potential combinatorial effects by drug treatments. We observed remarkable hematopoietic adaptability as the most primitive CD48^−^ HSCs decreased in numbers but increased in terms of proliferation at the peak of infection, consistent with reports studying other infection models [16,26,37,38]. MPP populations dramatically increased in size and proliferation while oligopotent progenitors virtually disappeared at the peak of the infection but their proliferation was never altered. This indicates that primitive hematopoietic populations adopt very different dynamics in response to a natural infection. The recovery of population size and proliferation within the hematopoietic system occurred within one month p.i., a clear indication of the ability of the organism to cope with such infection, which mice and humans have evolved with.

Malaria has been widely associated with a switch towards myelopoiesis at the expense of lymphopoiesis as innate immune cells are key in the efficient destruction of parasitized RBCs [30,39]. This was further supported by our data as we observed a deeper effect on the CMP population, which showed both the most dramatic decrease and slowest recovery compared to the other oligopotent precursors. Both the dramatic decrease in and slow recovery of the population size may have been results of the rapid induction and likely persistence of the fast differentiation of any cells reaching the CMP compartment. Interestingly, while all the oligopotent progenitors were depleted, the CD34^−^CD16/32^+^ population was shown to increase in numbers during acute infection, suggesting a possible compensating mechanism for the loss of committed progenitors. Given that this increase occurred without a corresponding rise in proliferation, it is unlikely that the CD34^−^ CD16/32^+^ cells amplified committed progenitors’ numbers during a phase of high consumption. It is likely instead that these cells resulted from the quick differentiation of expanding MPPs towards granulocyte and monocyte progenitors, consistent with accelerated differentiation and emergency granulopoiesis triggered by INF-γ [26]. By day 15, this same population may have additionally driven the rapid recovery of MEPs, which likely reflected the evolutionary requirement to restore healthy hematopoiesis as quickly as possible following infection. It has been described that at times of high demand for platelets, primitive hematopoietic cells can rapidly and directly differentiate into the megakaryocytic lineage, and it is possible that a similar process is activated during emergency granulopoiesis [40]. However, the different rates of recovery of different oligopotent progenitor populations may constitute a possible cause of some immunological abnormalities detected for a number of weeks in malaria survivors [41,42]. Moreover, we did not detect a change in the proliferation rate in any of the oligopotent progenitors, apart from a moderate increase in the proliferation of GMPs at day 11 p.i. In homeostasis, oligopotent progenitors had the highest proliferative rate, consistent with a loss of self-renewal and increased lineage commitment being associated with increased proliferation, leading to downstream ‘bursts’ of differentiated cell production [43,44,45]. The overall lack of changes in the proliferation rates of oligopotent progenitors is not limited to *Plasmodium* infection. Walter et al. [46] previously demonstrated that oligopotent progenitors were unresponsive to polyinosinic:polycytidylic acid (pI:pC) and that they were associated with a higher proliferative index during homeostasis compared to LT-HSCs. Oligopotent progenitors might already proliferate at their maximum capacity during homeostasis, which was supported by our study as we reported percentages of proliferating cells within mature progenitors to reach 40% and even 60%, while in the primitive compartments, they were below 20%. Alternatively, *Plasmodium*-induced stress might not be sufficient to elicit the proliferation of these cells, which might have a higher threshold of activation or only respond to different types of stress altogether. Altogether, our data suggest that oligopotent progenitors’ compartments are not replenished by their own proliferation but likely by the differentiation of upstream MPPs.

The MPP compartment was undoubtedly the one most remodeled by *Plasmodium* infection. It has been shown that upon the stimulation of emergency myelopoiesis, instructive differentiation signals are activated within the primitive compartment and lead to the expansion and skewing of the MPP compartment [31,47,48]. Consistent with the literature, our data showed a massive expansion of MPP2 and MPP3 populations, which was then recovered within day 24 p.i. This expansion was supported by dramatic increases in the proliferation rates of most MPP populations, especially the MPP5/6 population, which is hierarchically located between HSCs and MPP2-4 [32]. Despite the dramatic boost in proliferation, the numbers of MPP5/6 remained unchanged, underscoring the importance of this population in the production of the downstream MPPs, with cells transitioning very quickly through this compartment. On the contrary, the proliferation rate of megakaryocytic/erythroid-biased MPP2s remained relatively stable throughout our analysis, and at similar levels of oligopotent progenitors’ proliferation. It has been reported that following acute physiologic insult, the skewing of hematopoiesis into the production of myeloid cells leads to the impairment of erythro-/megakaryopoiesis [49]. Specifically, the inflammation-induced release of IL-1**β** or IFN-**γ** has been shown to directly increase the expression of Pu.1 on MPPs, which, in turn, stimulates myeloid production while inhibiting erythroid development [17,50]. It is possible that, in infected mice, MPP2 cells were not stimulated to proliferate further, and that the enlargement of the population may have resulted from impaired differentiation into downstream populations. Additionally, it is likely that, during malaria, MPP2 cells would be transcriptionally rewired to switch to a myeloid output, as was previously described for IL-7R^+^ c-Kit^hi^ lymphoid progenitors [25]. Similarly, it is expected that the transcriptional rewiring of lymphoid committed progenitors likely already initiates in the MPP4 population. Importantly, all primitive populations, including CD48^low/−^ HSPCs, MPP5/6, and MPP2, 3, and 4 returned to homeostatic size and proliferative rates by day 15 or 24 p.i., consistent with primitive progenitor populations responding directly to inflammatory stimuli rather than purely responding to a domino effect from the consumption of differentiated cells [9].

Analysis of the most primitive hematopoietic stem cells, CD48^−^ HSCs, revealed an initial depletion of the population, which had been previously demonstrated in response to *P. berghei* [26]. Many factors could explain this consumption, and it is generally accepted that inflammation causes the premature differentiation of HSCs at the expense of self-renewal [10,17]. The proportion of proliferating CD48^−^ HSCs increased during acute *P. chabaudi* infection, consistent with previous work with *P. berghei* [7] and other models of inflammatory stress [6,12,51,52]. The increased HSC proliferation likely feeds the expanding MPP populations. Interestingly, while, in chronic stresses, HSC depletion is long-lasting [16], in our recovery model, CD48^−^ HSC numbers were, on average, back to homeostatic levels by day 24 p.i.; however, they remained highly variable compared to those of all other primitive populations. Many mechanisms are in place to control the proliferation and differentiation of HSPCs, including feedback signals from neighboring cells or from the specialized HSC niche [53]. In addition, HSCs can sense the presence of infectious agents directly, and human *Plasmodium* species have been shown to take residence in the BM [23]. It is possible that the observed variable numbers of CD48^−^ HSCs are sentinels of a similar re-localization by *P. chabaudi*. Consistent with this, we observed a trend towards a slight increase in the numbers of all primitive HSPC populations at the day 29 p.i. time point, when a small recrudescence of *P. chabaudi* growth was noticed [28].

Our study highlighted the differences caused by acute versus chronic infection. Importantly, while both acute and chronic infections lead to similar consumption levels of HSPCs, emergency myelopoiesis, and proliferation within the system, the differences lie in the recovery stage. While acute infection only leads to a transient effect on HSPCs, which is then followed by the restoration of normal differentiation patterns, in chronic infections, hematopoietic disruption is typically long-lasting, leading to the sustained depletion of HSCs, HSPC exhaustion, and permanent differentiation skewed towards the myeloid lineage [6,15,16,37,54]. Moreover, the recovery of HSC numbers following their decrease at the peak of infections highlights the robustness of the system; however, it raises the question of whether the recovered HSC population may be less heterogeneous and/or less polyclonal than that of mice that remain healthy. This has important implications for the potential development of clonal hematopoiesis following severe or repeated infections. This would be highly relevant for the human population of endemic malaria regions, which could be at higher risk of developing hematological malignancies such as myeloid leukemia. It has already been shown that DNMT3a mutant HSCs have a selective advantage in surviving high levels of IFN-**γ** [15], a key upregulated cytokine in malaria infection [26,30,55,56], but repeated, infection-induced bottlenecks in the HSC population may lead to a reduction in HSC clonality even independently of pre-leukemic mutations, and this would inherently affect the robustness of the overall hematopoietic system in the long term.

In conclusion, our study demonstrated the high heterogeneity present within the hematopoietic system and outlined the behaviors specific to each hematopoietic stem and progenitor cell population in response to an infection. Importantly, we identified a window of time following *P. chabaudi* infection when recovery could be improved by boosting the hematopoietic restoration process. For instance, the targeted stimulation of MPPs to produce erythroid precursors could rescue impaired erythropoiesis, leading to the amelioration of severe anemia. Additionally, enhancing mature progenitors’ proliferation could reduce their depletion. Crucially, accelerating hematopoietic recovery would improve the return of immune and red blood cells to homeostatic levels and could minimize the risk of co-infection, which is often observed in malaria patients [57]. While we focused on phenotypically defined HSPC populations, future studies assessing the functionality of these cells, ideally including single-cell transcriptomics and functional assays, may further refine the dynamics of hematopoietic response to and recovery from acute infection and will be necessary to identify potential targets for interventions. Our findings highlight the significance of gaining a deeper understanding of these mechanisms to prevent the long-term effects associated with this and other infections.

## 4. Materials and Methods

### 4.1. Animals

All animal work, including animal care and experimental procedures, was carried out at Imperial College London (ICL) and at the Francis Crick Institute (FCI) in accordance with the current UK Home Office regulations (ASPA 1986) under license number PP9504146. C57BL/6 wild-type (WT) mice were bred and housed at the FCI. Animals were kept on Datesand Eco Pure Chips sawdust, Bed ‘r’ Nest nesting and smart homes enrichment, with Teklad Global Rodent Diet Sterilized, 18% Protein (catalogue number 2018S, Inotiv, West Lafayette, IN, USA). Female mice aged 6–10 weeks were used for all experimental procedures as *P. chabaudi* infection has been reported to cause more severe complications in male C57BL/6 mice that compromise welfare.

### 4.2. P. chabaudi Experimental Model

Cloned lines of *Plasmodium chabaudi chabaudi* (*AS* or *CB*) were used to initiate infections by intraperitoneal injection (i.p.) of 1 × 10^5^ infected RBCs collected from infected donor mice as previously described [58]. Blood-stage parasitemia was monitored by light microscopy assessment of Giemsa-stained (48900-500ML-F, Sigma-Aldrich, St. Louis, MO, USA) thin blood films generated from small drops of blood collected from the tail veins on the following days: 4, 7, 8, 10, 11, 14, 17, 21, 24, 29, 35, 40, 45, 50, 55, and 60 post infection (p.i.). Images of Giemsa-stained slides were acquired with the widefield microscope Zeiss Axio Observer, equipped with an IC5 color camera (AxioCam, Zeiss, Oberkochen, Germany), using a 63× oil objective. The average parasitemia was measured as a percentage of at least 1000 RBCs counted per slide. Mice were sacrificed at the time points of interest up to two months p.i. The number of mice analyzed at each time point is indicated in each figure legend.

### 4.3. Blood Cell Count Analysis

Blood was collected via cardiac puncture at the time of sacrifice and transferred in ethylenediaminetetraacetic acid (EDTA)-coated tubes. To determine the total count of WBCs, RBCs, and platelets, the blood was first diluted (1 in 5) in phosphate-buffered saline (PBS) and then run through the automatic hematology analyzer Sysmex XP-100 (Sysmex UK, Milton Keynes, UK).

### 4.4. Enzyme-Linked Immunosorbent Assay (ELISA)

Blood obtained by cardiac puncture was incubated at 4 °C for at least 3 h to allow clotting and subsequently centrifuged at 12,000× *g* for 10 min at 4 °C. The serum supernatant was collected and stored at −20 °C until required. IFN-γ and TNF-α ELISAs (catalogue numbers, 430104 and 430904, BioLegend, San Diego, CA, USA) were performed using the serum according to the manufacturer’s instructions and by diluting samples 1:2.

### 4.5. EdU Incorporation Assay

For cell populations’ proliferation analysis, 5-ethynyl-2′-deoxyuridine (EdU) was administered 2.5 h prior to culling via intravenous injection (1 mg/mouse). EdU, a thymidine analog, was incorporated in newly synthesized DNA, thus labeling the cells in the S-phase of the cell cycle. EdU uptake was measured by flow cytometry (see next section).

### 4.6. Flow Cytometry

For the phenotypic analysis of HSPCs during infection, femurs, tibias, and hips were harvested at day 8, day 11, day 15, day 24, day 29, and day 60 p.i. from age-matched healthy control and infected mice. Bones were crushed in FACS buffer (PBS, 2% fetal bovine serum (FBS)) and obtained cells were filtered through 40 µm strainers, depleted of RBCs, and stained with relevant monoclonal antibodies. For a comprehensive summary of antibody concentrations, clones, and manufacturing companies, consult Appendix A. Dead cells were excluded by staining the cells with fixable viability stain 510 (BD Horizon).

For EdU detection, the Click-iT™ EdU Pacific Blue™ Kit (catalogue number, C10418, Invitrogen, Carlsbad, CA, USA) was employed according to the manufacturer’s instructions. Briefly, single-cell suspensions were stained with the relevant antibodies, then fixed with 4% paraformaldehyde (PFA) and permeabilized, followed by incubation with the Click-iT™ detection cocktail.

To determine the absolute cell number in the populations of interest, Sysmex XP-100 or Calibrite beads (catalogue number 340486, BD Biosciences, Macquarie Park, NSW, Australia) were used. For the former, the number of WBCs in two legs was multiplied by the frequency of single cells obtained via flow cytometry analysis. For the latter, Calibrite beads were added to the suspension prior to running each sample through the flow cytometer. Absolute numbers were then calculated by multiplying the beads added (100,000) by the cell count of each population and divided by the number of beads detected by the machine.

Relevant controls, such as fluorescent minus one (FMO) and single-color controls, were included for compensation and gating purposes. All samples were analyzed on an LSR Fortessa (BD Biosciences) III, and the software FlowJo (version 10.10.0, Tree Star) was used to analyze the resulting data.

### 4.7. Statistical Analysis

Raw data were tabulated in Microsoft Excel while graphs and statistical tests were created and performed via GraphPad Prism (version 10.10.0). Data are expressed as mean +/− standard error of the mean (s.e.m.) values. Where stated, absolute cell numbers were normalized by dividing each value by the average of the corresponding control value for that day. For multiple comparisons, unpaired two-tailed *t*-test with Holm–Šidák correction was used [59]. Differences were considered significant where (*) *p* < 0.05, (**) *p* < 0.01, (***) *p* < 0.001, or (****) *p* < 0.0001. Detailed statistical information can be found in each figure caption. For all experiments, *n* refers to the total number of replicates at each time point, pooled from three independent experiments.

## Figures and Tables

**Figure 1 ijms-26-02816-f001:**
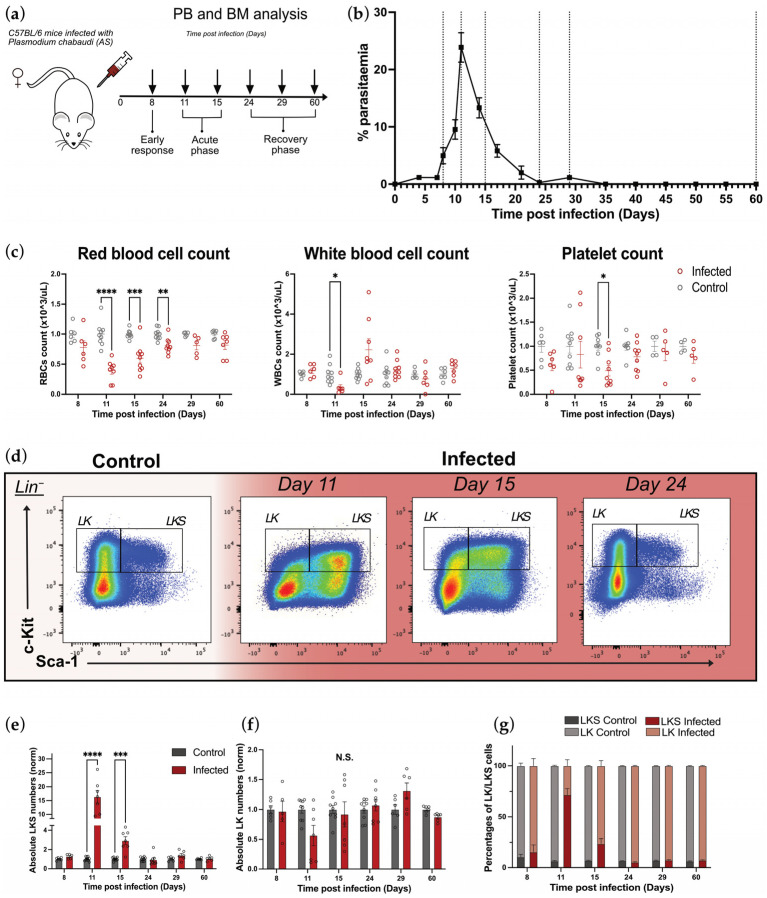
*P. chabaudi* is an effective model for studying perturbed hematopoiesis and its recovery. (**a**) Experimental set-up. On day 0, C57BL/6 mice were injected with *P. chabaudi AS*-infected RBCs and PB and BM analyses were conducted at days 8, 11, 15, 24, 29, and 60 p.i. (**b**) Percentages of infected RBCs in PB over the course of the infection. Dotted lines highlight days when PB and BM were analyzed. (**c**) PB cell counts in control and infected mice during the infection, including RBC, WBC, and platelet counts. (**d**) Representative flow cytometry pseudo-color density plots showing LKS and LK populations in control and infected mice at days 11, 15, and 24 p.i. (**e**,**f**) Normalized absolute numbers of LKS (**e**) and LK (**f**) cells in control and infected mice at days 8, 11, 15, 24, 29, and 60 p.i. (**g**) Normalized percentages of LKS and LK cells within the Lin^−^ c-Kit^+^ population in control and infected mice at days 8, 11, 15, 24, 29, and 60 p.i. In (**b**,**c**,**e**–**g**), data are presented as mean +/− s.e.m. values, and *p* values (asterisks) were determined by unpaired two-tailed Student’s *t*-test. * *p* < 0.05, ** *p* < 0.01, *** *p* < 0.001, **** *p* < 0.0001, N.S. non-significant. *n* = 6, 9, 9, 9, 6, and 6 for control and *n* = 5, 7, 7, 8, 6, and 6 for infected mice at days 8, 11, 15, 24, 29, and 60 p.i., respectively, pooled from three independent infections. Abbreviations: PB, peripheral blood; BM, bone marrow; RBC, red blood cell; WBC, white blood cell; LK, Lineage^−^cKit^+^Sca-1^−^; LKS, Lineage^−^cKit^+^Sca-1^+^.

**Figure 2 ijms-26-02816-f002:**
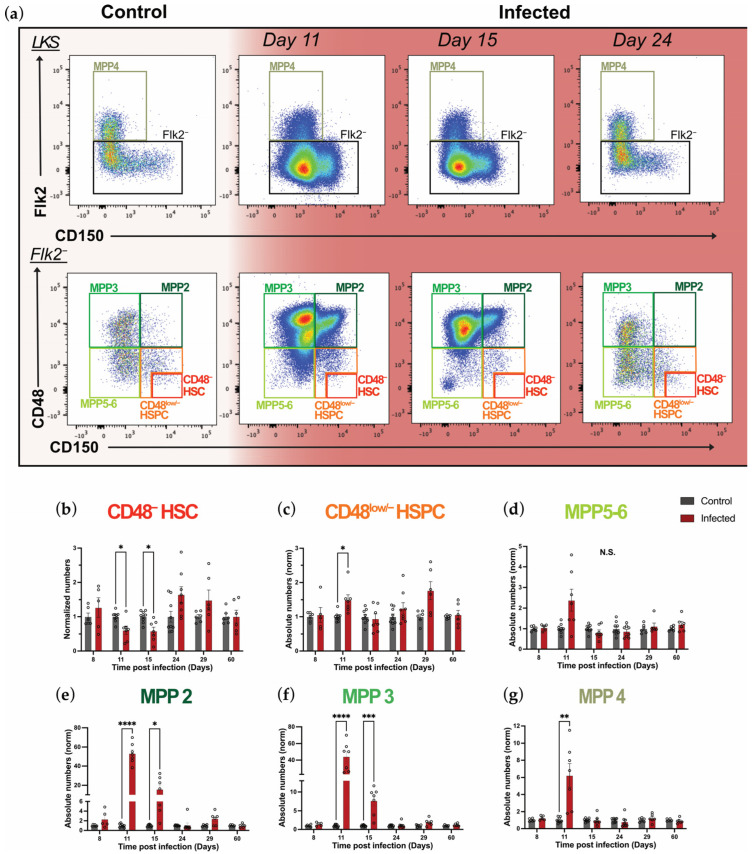
*P. chabaudi* infection effects on primitive hematopoietic populations are resolved upon pathogen clearance. (**a**) Representative flow cytometry plots showing primitive HSPC populations in control and infected mice at days 11, 15, and 24 p.i. (**b**–**g**) Normalized absolute numbers of CD48^−^ HSC (**b**), CD48^low/−^ HSPC (**c**), MPP5-6 (**d**), MPP2 (**e**), MPP3 (**f**), and MPP4 (**g**) cell populations in control and infected mice at days 8, 11, 15, 24, 29, and 60 p.i. In (**b**–**g**), data are presented as mean +/− s.e.m. values, and *p* values (asterisks) were determined by unpaired two-tailed Student’s *t*-test. * *p* < 0.05, ** *p* < 0.01, *** *p* < 0.001, **** *p* < 0.0001, N.S. non-significant. *n* = 6, 9, 9, 9, 6, and 6 for control and *n* = 5, 7, 7, 8, 6, and 6 for infected mice at days 8, 11, 15, 24, 29, and 60 p.i., respectively, pooled from three independent infections. Abbreviations: HSC, hematopoietic stem cell; HSPC, hematopoietic stem and progenitor cell; MPP, multipotent progenitor.

**Figure 3 ijms-26-02816-f003:**
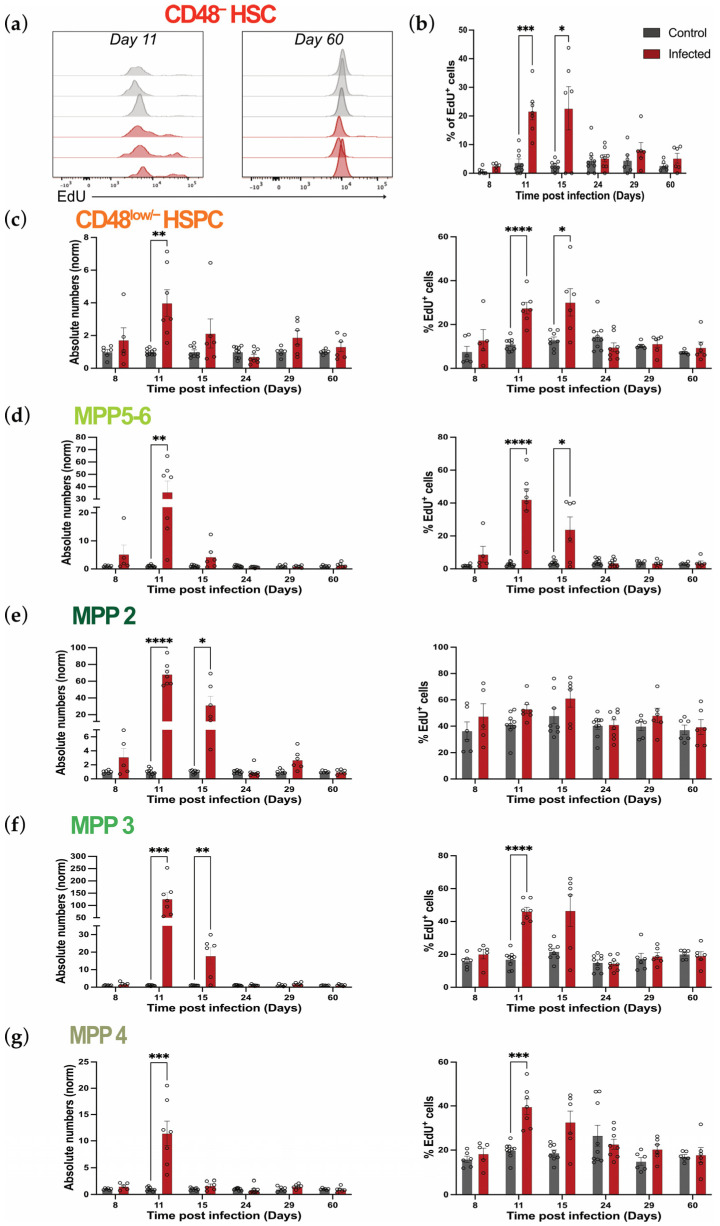
Proliferation dynamics of primitive HSPC populations during *P. chabaudi* infection. (**a**) Representative flow cytometry data histograms showing EdU incorporation in CD48^−^ HSCs in control and infected mice at day 11 and 60 p.i. (each histogram represents one mouse; plots are representative of all time points analyzed from 3 independent infections). (**b**) Percentages of EdU^+^ cells in CD48^−^ HSCs in control and infected mice at days 8, 11, 15, 24, 29, and 60 p.i. (**c**–**g**) Normalized absolute numbers (left) and percentages (right) of EdU^+^ cells within CD48^low/−^ HSPC (**c**), MPP5-6 (**d**), MPP2 (**e**), MPP3 (**f**), and MPP4 (**g**) populations in control and infected mice at days 8, 11, 15, 24, 29, and 60 p.i. In (**b**–**g**), data are presented as mean +/− s.e.m. values, and *p* values (asterisks) were determined by unpaired two-tailed Student’s *t*-test. * *p* < 0.05, ** *p* < 0.01, *** *p* < 0.001, **** *p* < 0.0001. Data were pooled from three independent infections. *n* = 6, 9, 9, 9, 6, and 6 for control and *n* = 5, 7, 7, 8, 6, and 6 for infected mice at days 8, 11, 15, 24, 29, and 60 p.i., respectively.

**Figure 4 ijms-26-02816-f004:**
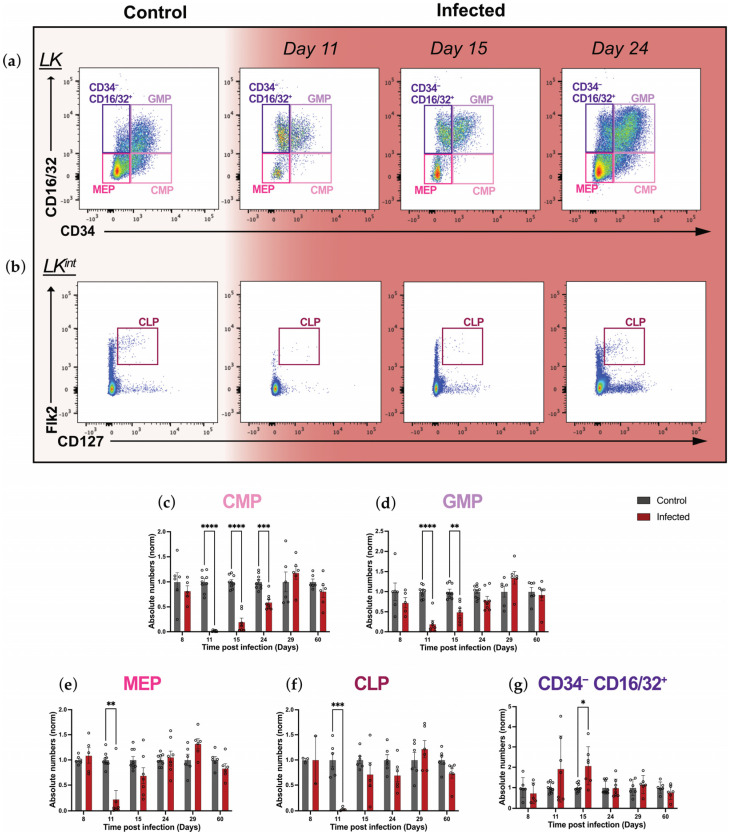
Mature progenitors are largely lost during acute *P. chabaudi* infection. (**a**,**b**) Representative flow cytometry plots showing CD34^−^CD16/32^+^, GMP, CMP, and MEP LK subpopulations (**a**) and CLPs (**b**) in control and infected mice at days 11, 15, and 24 p.i. (**c**–**g**) Normalized absolute numbers of CMP (**c**), GMP (**d**), MEP (**e**), CLP (**f**), and CD34^−^CD16/32^+^ (**g**) progenitors in control and infected mice at days 8, 11, 15, 24, 29, and 60 p.i. In (**c**–**g**), data are presented as mean +/− s.e.m. values, and *p* values (asterisks) were determined by unpaired two-tailed Student’s *t*-test. * *p* < 0.05, ** *p* < 0.01, *** *p* < 0.001, **** *p* < 0.0001. *n* = 6, 9, 9, 9, 6, and 6 for control and *n* = 5, 7, 7, 8, 6, and 6 for infected mice at days 8, 11, 15, 24, 29, and 60 p.i., respectively. Abbreviations: LK^int^, Lineage^−^ c-Kit^intermediate^; CMP, common myeloid progenitor; GMP, granulocyte–macrophage progenitor; MEP, megakaryocyte–erythrocyte progenitor; CLP, common lymphoid progenitor.

**Figure 5 ijms-26-02816-f005:**
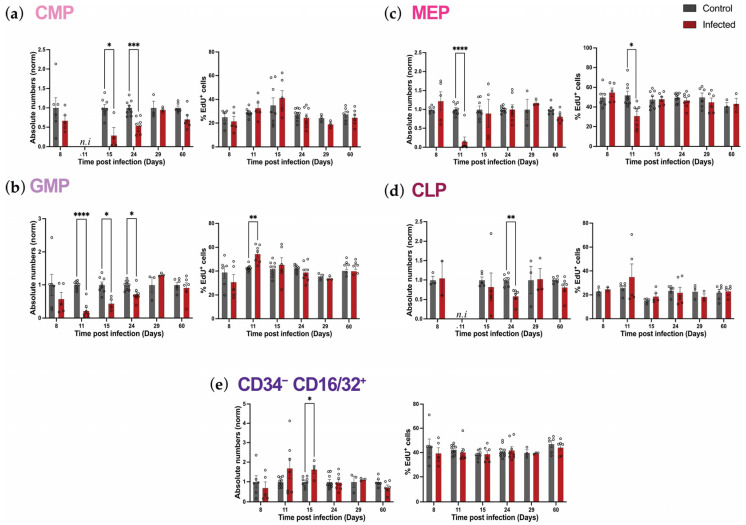
*P. chabaudi* infection does not alter the proliferation of mature progenitors. (**a**–**e**) Normalized absolute numbers (left) and percentages (right) of EdU^+^ cells in CMP (**a**), GMP (**b**), MEP (**c**), CLP (**d**), and CD34^−^ CD16/32^+^ (**e**) populations in control and infected mice at days 8, 11, 15, 24, 29, and 60 p.i. All data are presented as mean +/− s.e.m. values, and *p* values (asterisks) were determined by unpaired two-tailed Student’s *t*-test. * *p* < 0.05, ** *p* < 0.01, *** *p* < 0.001, *p* **** < 0.0001. Data were pooled from three independent infection experiments. *n* = 6, 9, 9, 9, 6, 6 for control and *n* = 5, 7, 7, 8, 6, 6 for infected mice at days 8, 11, 15, 24, 29, and 60 p.i., respectively. n.i.: not included.

**Figure 6 ijms-26-02816-f006:**
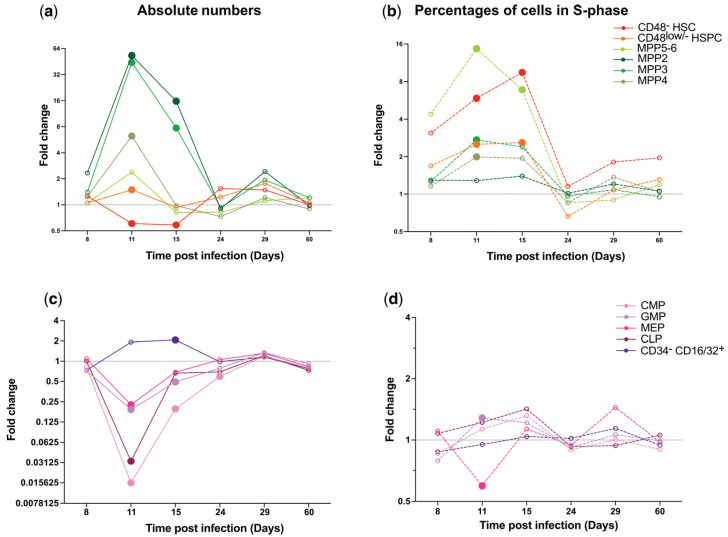
Hematopoietic populations adopt very different responses when responding to and recovering from severe infection. (**a**) Average fold change of normalized absolute numbers of primitive hematopoietic cell populations. (**b**) Average fold change of percentages of EdU^+^ cells in primitive HSPC populations. (**c**) Average fold change of normalized absolute numbers of oligopotent progenitor populations. (**d**) Average fold change of percentages of EdU^+^ cells in oligopotent progenitor populations. Data are transformations of data shown in Figure 2, Figure 3, Figure 4 and Figure 5. 1 = average control values; values between 0 and 1 represent a decrease; values > 1 represent an increase. Statistically significant differences between the infected and control groups are indicated by bigger, filled circles. Non-statistically significant differences are indicated by smaller, empty circles. Data were pooled from three independent infection experiments. *n* = 6, 9, 9, 9, 6, and 6 for control and *n* = 5, 7, 7, 8, 6, and 6 for infected mice at days 8, 11, 15, 24, 29, and 60 p.i., respectively.

## Data Availability

The data presented in this study are openly available as Zenodo data at https://doi.org/10.5281/zenodo.15019058 [60].

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
