# Peer review of "Differential Response and Recovery Dynamics of HSPC Populations Following Plasmodium chabaudi Infection"

_ijms, 2025, doi:10.3390/ijms26062816_

Round 1
Reviewer 1 Report
Comments and Suggestions for Authors
Bruno et al. report the use of a time course experimental approach to investigate the effects of P. chabaudi infection on primitive hematopoietic stem and progenitor cells (HSPCs) and oligopotent progenitors in the bone marrow throughout the acute and recovery stages in infected mice.
1. Although defined changes are shown in acute infection compared to the recovery phase in the P. chabaudi model, it is unclear if these changes would be reproduced in other rodent Plasmodium infections. In models with a more pronounced acute stage of infection and inflammatory response, the expansion of the LSK compartment as shown in Figures 1D may have a more skewed and heterogeneous MPP and HSC profile. The authors allude to this in section 2.2. Additional explanation of the differences between cell consumption, recovery and cellular homeostasis in inflammatory models and chronic infection models should be included in the discussion section.
2. In sections 2.4 and 2.5, and Figure 4A, what was the corresponding parasite profile and parasitemia as CMP, GMP, MEP and CLP populations decreased? Similarly, how was the cytokine profile affected in the mice?
3. Include Giemsa-stained P. chabaudi infected erythrocytes at days 8, 11,15, 24, 29, and 60 p. i.
4. Was compensation (cell proliferation and differentiation) observed from the CD34-CD16/32+ population during the slow recovery of other oligopotent progenitors?
5. The discussion should be reorganized and shortened with a focus on how the data obtained regarding proliferation, differentiation and recovery of HSC and MMP populations can inform the understanding of cellular dynamics in human malaria.
Lines 11 and 12: “…how the host responds to severe infection.”
Lines 66-69: indicate “entrance” of parasite into the skin is through the bite of the female Anopheles mosquito. Specify the stages of Plasmodium entering the liver, found within the liver and emerging from the liver.
Italicize all genus and species names.
Comments on the Quality of English LanguageMinor corrections in English required for clarity of the manuscript content.
Reviewer 2 Report
Comments and Suggestions for Authors
The article presents an intriguing study on hematopoietic stem and progenitor cells (HSPCs, responsible for the renewal and maintenance of blood cells) during the acute phase and recovery upon pathogen clearance in Plasmodium chabaudi infection. The introduction is well-written and provides a clear context for readers, even those not specialized in the field. The approach of the article involves the analysis of two groups of C57BL/6 mice (control and infected with Plasmodium chabaudi), which were examined through analysis of peripheral blood and bone marrow cells at different timepoints following infection: day 8 (early response, pre-parasitemia peak), days 11 and 15 (acute phase, peak and post-peak parasitemia, respectively), and days 24, 29, and 60 (recovery phase, post-peak, with very low to undetectable parasitemia).
In this context, I would suggest that the authors provide further clarification on the diet administered to the mice during this period, as diet has a direct influence on the induction of pro-inflammatory or anti-inflammatory cytokines, potentially affecting the number of immune cells. The methodology used in the analyses (sections 4.3, 4.4, and 4.5) is complementary and provides robust and interesting results. However, I was unable to find information in the manuscript regarding the number of mice used in the study. The statistical analysis was correctly applied, and the conclusions drawn are consistent with the results obtained.
Round 2
Reviewer 1 Report
Comments and Suggestions for Authors
The authors have answered all queries.
Author Response
Thank you